# Preparation and Application of Quaternized Chitosan- and AgNPs-Base Synergistic Antibacterial Hydrogel for Burn Wound Healing

**DOI:** 10.3390/molecules26134037

**Published:** 2021-07-01

**Authors:** Xushan Chen, Huimin Zhang, Xin Yang, Wuhong Zhang, Ming Jiang, Ting Wen, Jie Wang, Rui Guo, Hanjiao Liu

**Affiliations:** 1Shenzhen Bao’an Traditional Chinese Medicine Hospital Group, Shenzhen 518133, China; chriscachan@163.com (X.C.); huiminzhang000@163.com (H.Z.); zhangwuhong000@163.com (W.Z.); 2Shenzhen Bao’an Traditional Chinese Medicine Hospital, Guangzhou University of Chinese Medicine, Shenzhen 518033, China; enjoyxin99@163.com; 3Key Laboratory of Biomaterials of Guangdong Higher Education Institutes, Engineering and Technological Research Centre for Drug Carrier Development, Department of Biomedical Engineering, Jinan University, Guangzhou 510632, China; mingjiang201901@163.com; 4Fujian University of Traditional Chinese Medicine, Fuzhou 350122, China; wenting2021521@163.com (T.W.); WJ1095235996@163.com (J.W.)

**Keywords:** infected burn wound, antibacterial hydrogel, silver nanoparticles

## Abstract

Infection is the major reason that people die from burns; however, traditional medical dressings such as gauze cannot restrain bacterial growth and enhance the healing process. Herein, an organic- and inorganic-base hydrogel with antibacterial activities was designed and prepared to treat burn wounds. Oxidized dextran (ODex) and adipic dihydrazide grafted hyaluronic acid (HA-ADH) were prepared, mixed with quaternized chitosan (HACC) and silver nanoparticles to fabricate Ag@ODex/HA-ADH/HACC hydrogel. The hydrogel, composed of nature biomaterials, has a good cytocompatibility and biodegradability. Moreover, the hydrogel has an excellent antibacterial ability and presents fast healing for burn wounds compared with commercial Ag dressings. The Ag@ODex/HA-ADH/HACC hydrogel will be a promising wound dressing to repair burn wounds and will significantly decrease the possibility of bacterial infection.

## 1. Introduction

The skin, one of the largest and important organs, is the natural barrier of the human body [1]. It can restrain water loss, electrolytes and nutrients in the body, and protect the organs and tissues of the body from external mechanical, physical and chemical stimulation [2,3]. Wound healing is an intricated process including several complex biological phases including inflammatory, proliferation and reconstruction [4]. Once the skin is traumatized by external damage, some minor wounds can heal by themselves, but when the wound is severe such as in the case of burns, the skin may lose its ability to repair itself and, in severe cases, may cause death [5]. Therefore, the development of novel materials that prevent infections from burn wounds is paramount.

Since the early 1980s, wound dressings, which have the properties of promoting skin healing and protecting wound areas from infection, have been gradually studied and applied in clinical fields [6]. Traditional wound dressings mainly include dry gauze and oil yarn, which can only cover the wound. However, due to their possible secondary injury to the wound during dressing change, their poor air permeability and the low absorption rate of exudate, etc., they are not the ideal wound dressing thus far. Hydrogel has been used to mimic an extracellular matrix for a long time. As a new type of biomedical dressing, a hydrogel dressing has many adjustable physical and chemical properties. It is easy to functionalize by changing materials and can be used as a drug carrier [7]. Antibacterial hydrogel dressings fall into two broad categories by reason of the antibacterial mechanism. One type is the hydrogel with an inherent antimicrobial activity. These hydrogels, such as peptide-based hydrogels [8,9], chitosan-containing hydrogels [10,11,12] and other polymer-derived hydrogels, themselves can defend against microorganisms. Another type is the hydrogel combined with antibacterial agents such as antibiotics [13] and inorganic nanomaterials [14,15,16]. However, studies investigating the application of the synergistic antimicrobial hydrogels with inherent antimicrobial activity and antibacterial agents in wound healing is limited. Additionally, such antibacterial hydrogel will be of great significance to the development of burn wound dressing.

Previous studies revealed that chitosan-based hydrogels could be used in wound healing by reason of its inherent antibacterial properties [17,18]. It can absorb exudates and prevent the microorganisms of the external environment invading the wound through forming a thin barrier [19]. The mechanism of the antibacterial activity of chitosan is by reason of the positive charge. Chitosan can adsorb negatively charged bacteria and further accumulate on the cell wall through electrostatic and hydrophobic interactions with bacteria, then disrupt the bacterial membrane and cause cytoplasmic leakage, finally resulting in the lysis and death of the bacteria [11,20]. Nevertheless, the antibacterial activity of chitosan is restricted by its solubleness because few can dissolve when the pH value is greater than 6.5 [21]. A chitosan derivative (chitosan quaternary ammonium salt) can be obtained by replacing the amino group of chitosan with a quaternary amino group, which improves its inherent properties, such as its antibacterial capacity, solubility, adsorption and moisture retention [22,23]. Therefore, it has a significantly improved application range in comparison to chitosan.

Diverse types of inorganic particles such as metal, metal oxides, non-metal oxides and others have been applied as an antibacterial agent. Ag nanoparticles (AgNPs) is one of the inorganic antibacterial agents that is extensively used in biomedical areas [15]. AgNPs have an extensive spectrum of biocidal and antibacterial activities, mainly because they can damage the respiratory system and transmission system in bacterial cells, thereby inhibiting the bacteria’s metabolism and hindering the reproduction of the bacteria’s DNA [24,25]. Furthermore, they can also disrupt the scavenging mechanisms by directly binding with thiol groups in related enzymes and turn GSH into its oxidized form, glutathione disulfide (GSSG), which results in increased concentrations of ROS and free radicals, and further direct damage to the bacteria [24]. In addition, AgNPs not only have antibacterial properties, but also have an anti-inflammatory ability, because they can prevent pro-inflammatory factors such as TNF-α, IL-6 and IL-1β from expressing [26]. Thereby, the AgNPs could be a desired candidate for facilitate burn wound healing. Previous studies have shown that quaternary ammonium chitosan and silver nanoparticles have synergistic antibacterial effects [27], but the hydrogels that combine with these two complexes are rarely reported.

In this study, the organic- and inorganic-base hydrogel with antibacterial activities was designed and prepared. The preparation and application of the antibacterial hydrogels are shown in Scheme 1. In brief, oxidized dextran (ODex) and adipic dihydrazide grafted hyaluronic acid (HA-ADH) were prepared, mixed with quaternized chitosan (HACC) and Ag nanoparticles to fabricate the Ag@ODex/HA-ADH/HACC hydrogel. Histological and Western blot analyses revealed that Ag@ODex/HA-ADH/HACC exhibited good anti-inflammatory effects, and the skin displayed a well-organized collagen deposition that facilitated burn wound healing. To sum up, the present results revealed that an Ag@ODex/HA-ADH/HACC hydrogel with inherent antimicrobial activity and biocompatibility can facilitate burn wound healing.

## 2. Results and Discussions

### 2.1. Characterization of the Chemical Construction and Morphology of the Hydrogel

The characterizations of FTIR, shown in Figure 1A, proved that the dextran and hyaluronic acid had been successfully modified. The characteristic peak of HA-ADH at 1610 cm^−1^ was associated with the variable-angle vibration of -NH_2_, confirming that ADH was successfully grafted with HA [28]. Moreover, the absorption peak that appeared at 1729 cm^−1^ of the ODex was attributed to the vibration of the aldehyde groups, indicating that the dextran had been oxidized by sodium periodate [29,30]. Furthermore, the characteristic peak at 1670 cm^−1^ that presented in all hydrogels was by reason of the hydrazone bonds formed between ODex and HA-ADH via the Schiff base [31]. In Figure 1B, a distinct absorption peak was discovered at ~417 nm, which contributed to the plasmon resonance absorption peak of the AgNPs [32], demonstrating that silver nitrate was reduced to AgNPs by chitosan successfully, and loaded onto the hydrogel.

Generally, the smaller the size of the AgNPs is, the better the antibacterial effect is [33]. Appendix A exhibits the size of the hydrogel without silver nanoparticles, and the mean size of sparingly soluble polysaccharides was greater than 190 nm and within the scope of 190–550 nm. The size of the AgNPs in Figure 1C, detected using dynamic light scattering (DLS), were mostly (>50%) within the scope of 60–200 nm, which could not be observed in Appendix A. Thus, we could infer that the silver nanoparticles were within the size distribution of 60–190 nm. Suitable porosity at a range of 60–90% permits the wound to obtain nutrients, facilitate cell respiration and penetration, and maintain a moist environment on the wound’s surface [34]. In addition, previous research has proven that a mean pore size of about 30–40 μm can promote the proliferation of fibroblasts [35]. From Figure 1D, the pore size distribution of the ODex/HA-ADH, ODex/HA-ADH/HACC and Ag@ODex/HA-ADH/HACC hydrogels was 30–110, 20–75 and 25–110 μm, respectively. In addition, the porosity of the hydrogels was 84.9, 80.3 and 72.6%, respectively, indicating that the hydrogel we prepared was suitable for skin cell metabolism and wound regeneration. Furthermore, a partial enlargement of the SEM image of the Ag@ODex/HA-ADH/HACC hydrogel indicated that AgNPs were successfully loaded onto the hydrogel and the average diameter was about 50–100 nm. The size obtained from SEM was smaller than the previous result of DLS, possibly due to the different preparation of the sample: the realistic size observed using SEM was prepared in desiccative conditions, while the size detected using DLS was a hydrodynamic diameter [36].

### 2.2. Swelling, In Vitro Degradation and the Rheology Behaviors of Hydrogels

Wound exudate possibly leads to excess moisture in the wound, further influencing the wound to repair. The water uptake ability of hydrogel could be manifested by the swelling ratio, which could remove excess moisture to restrict infection [37]. In Figure 2A, after being immersed in the liquor for 80 h, the maximum water absorption of whole hydrogels attained 40 times their initial weight. The swelling rate of the hydrogel that was immerged in deionized water is higher than that of the hydrogels immersed in a PBS solution, which is due to the lower density of water than that of the PBS solution [38]. Moreover, the swelling rate of ODex/HA-ADH/HACC is higher than ODex/HA-ADH, which is attributed to the chitosan, as it has plentiful hydrophilic groups such as amine groups that enhanced swelling ratio largely [39].

The effect of the different composition on the rheological properties of the hydrogel was explored using the storage modulus (G’) and loss modulus (G’’) (Figure 2B). All three types of hydrogels could maintain a gel state with an increased amount of time. However, the Ag@ODex/HA-ADH/HACC hydrogel presented the lowest storage modulus (G’) among these three hydrogels. This may be owed to the fact that silver ions were reduced to AgNPs by chitosan, which reduced the crosslinking density of the hydrogel [40].

The suitable degradation rate of hydrogels is also a significant consideration for wound dressing [41]. In this study, we explored the degradation rates of ODex/HA-ADH and ODex/HA-ADH/HACC hydrogels in PBS (Figure 2C) and a hyaluronidase solution (Figure 2D), respectively. From the figure, the degradation rate of ODex/HA-ADH was 43% in PBS and 54% in the hyaluronidase solution after 15 days, while ODex/HA-ADH/HACC was 31 and 45%, respectively. The ODex/HA-ADH and ODex/HA-ADH/HACC hydrogels showed a higher degradation rate in the hyaluronidase solution, this is mainly because the hyaluronic acid in the hydrogels could be decomposed by hyaluronidase, which destroyed the main chain of hyaluronic acid, further cleaving the internal connection of N-acetyl-D-glucosamine. However, in the PBS liquor, the degradation is mainly based on the hydrolytic scission of molecular chains, and the broken amide bond and polysaccharide chain [42].

### 2.3. In Vitro Cytotoxicity of Hydrogel

L929 cells were used to evaluate the cytotoxicity of the hydrogels. The L929 cells cocultured with ODex/HA-ADH and ODex/HA-ADH/HACC hydrogel extracts showed a spindle shape, and the number of cells increased over time. In contrast, cells in the Ag@ODex/HA-ADH/HACC hydrogel became partially round, indicating that the cells were partially dead due to the cytotoxicity of the Ag nanoparticles (Figure 3A). After that, we used a CCK-8 assay to quantitatively analyze the cell viability (Figure 3B). By detecting the OD value at 450 nm, the ODex/HA-ADH and ODex/HA-ADH/HACC hydrogels had a higher cell viability (over 90%) compared with the Ag@ODex/HA-ADH/HACC hydrogel (76.41%) on day one. This may be due to the fact that dextran and hyaluronic acid were all-natural biomaterials with good biocompatibility that can promote cell proliferation [43,44], while AgNPs had a certain cytotoxicity [45]. Moreover, the cell viability on day seven was higher than on day one and day three, this may be attributed to the ODex/HA-ADH/HACC hydrogel promoting cell proliferation. Collectively, the results proved that all of the hydrogels had great biocompatibility, which may have potential application in burn wound healing.

### 2.4. Antibacterial Activity of Hydrogel In Vitro

Wounds exposed to the air are greatly susceptible to bacterial infections and a further delay to the wound healing process; therefore, the hydrogels with strong antibacterial properties that covered the wound completely have a greater clinical application value. *E. coli*, *S. aureus* and *P. aeruginosa*, the typical types of bacteria found in wounds, have been chosen to assess the antibacterial properties of the materials [46]. Figure 4A shows the inhibition zone pictures of the ODex/HA-ADH, ODex/HA-ADH/HACC and Ag@ODex/HA-ADH/HACC hydrogels. The ODex/HA-ADH hydrogel had no antibacterial properties, while the hydrogels ODex/HA-ADH/HACC and Ag@ODex/HA-ADH/HACC had inhibition zones in three kinds of bacteria. It may be due to the cationic group of HACC that increased the positive charge density of the hydrogel, which could adsorb the negatively charged bacteria and further accumulate on the cell wall through electrostatic and hydrophobic interactions with bacteria, then disrupt the bacterial membrane and cause cytoplasmic leakage, leading to the lysis and death of the bacteria [47]. Figure 4B shows the corresponding statistical data of the inhibition zone of *E. coli*, *S. aureus* and *P. aeruginosa*. The inhibition zone of the Ag@ODex/HA-ADH/HACC hydrogel was 24, 24 and 27 mm on *E. coli*, *S. aureus* and *P. aeruginosa*, respectively, while the ODex/HA-ADH/HACC hydrogel was 16, 20, 17 mm, respectively. The antibacterial ability of the Ag@ODex/HA-ADH/HACC hydrogel was significantly higher than the ODex/HA-ADH/HACC hydrogel, which can be attributed to the synergistic antibacterial properties of AgNPs and HACC. AgNPs could attach onto the bacteria wall and membrane, then damage the intracellular biomolecules and structures to kill the bacteria [24]. In conclusion, the antibacterial effect of Ag@ODex/HA-ADH/HACC was greater than the other group, indicating that Ag@ODex/HA-ADH/HACC has the potentiality to be used as an antibacterial wound dressing in clinics.

### 2.5. In Vivo Wound Healing Performance of Hydrogel

An infected burn wound rat model was used to further demonstrate the biocompatibility and healing efficiency of the Ag@ODex/HA-ADH/HACC hydrogel. On days three and seven post-surgery, obvious differences in the wound healing rates were found. As shown in Figure 5A,B, all the wounds on the third day were enlarged compared to the first day, this may be due to the ulceration caused by the infection. However, the Ag@ODex/HA-ADH/HACC hydrogel-treated group did not show obvious expansion, which may owe to the good antibacterial ability of the hydrogel. The wound area treated with hydrogels Ag@ODex/HA-ADH/HACC and ODex/HA-ADH/HACC reduced to 41.3 and 45.2% after 7 days, respectively, while the wound area in all other groups were still over 60%. On the 14th day after treatment, the wounds exposed to hydrogels ODex/HA-ADH/HACC and Ag@ODex/HA-ADH/HACC had almost completely healed, while more than 20% were still unhealed in the other groups. After 21 days of treatment, the wounds of all the groups were healed, while the Ag@ODex/HA-ADH/HACC-treated group presented the best healing effect and the smallest scars after healing. The content of bacteria, as shown in Figure 5C, revealed the wounds were successfully infected with *P. aeruginosa* and further indicated the Ag@ODex/HA-ADH/HACC group had excellent antibacterial activity in vivo.

To access the wound healing histologically, wound tissues were collected 3-, 7-, 14- and 21-days post-treatment, respectively. The wound tissues were exposed to H&E and Masson’s trichrome staining, which were further used to verify the therapeutic effects of the ODex/HA-ADH, ODex/HA-ADH/HACC and Ag@ODex/HA-ADH/HACC hydrogels and commercial silver-containing dressings (Aquacel^®^ Ag). H&E staining was used to analyze the inflammation, proliferation of fibroblasts and capillaries in the wound [48,49]. On the seventh day, all the groups could still observe the inflammatory cells that presented in the wound tissues. The ODex/HA-ADH and Aquacel^®^ Ag groups still had several inflammatory cells, while the ODex/HA-ADH/HACC and Ag@ODex/HA-ADH/HACC showed a small amount of epithelial tissue on the seventh day. On the 14th day, epithelial tissue had formed in all four groups. However, on the 14th day, the Ag@ODex/HA-ADH/HACC hydrogel-treated group had begun to form the dermal papilla and presented a new hair follicle. Furthermore, on the 21st day, the tissue morphology repaired by Ag@ODex/HA-ADH/HACC was similar to the skin tissue of healthy rats [50], indicating that Ag@ODex/HA-ADH/HACC possesses better effects compared to Aquacel^®^ Ag (Figure 6A).

The amount of new collagen deposition could be evaluated using Masson’s trichrome staining (Figure 6B). More mature and well-grown collagen depositions were presented in the Ag@ODex/HA-ADH/HACC group. In conclusion, the results showed that wounds treated with Ag@ODex/HA-ADH/HACC exhibited inconspicuous inflammation, presented almost complete re-epithelialization and exhibited well-organized collagen deposition.

### 2.6. Immunohistochemistry and Western Blot Analysis

Wound healing is deemed to be an intricate process; inflammatory cells could be induced by the expression of several proinflammatory cytokines, which enhance wound healing during the early wound repairing process and cause the response of local and systemic defense [51,52]. Neutrophils and monocytes can generate Interleukin-6 (IL-6), which participated in the regulation of leukocyte infiltration [53]. Furthermore, low concentrations of TNF-α and IL-1β could indirectly stimulate the inflammatory response and promote the production of growth factors by macrophages to enhance wound healing, while high levels could have an adverse influence on wound healing [54,55]. As shown in Figure 7, on the 14th day post-treatment, all the groups had different degrees of inflammation, and the group control without antibacterial effect presented significant expressions of IL-6, IL-1β and TNF-α. The ODex/HA-ADH/HACC and Aquacel Ag groups exhibited less expression than the group control; however, the expression of IL-6, IL-1β and TNF-α in the Ag@ODex/HA-ADH/HACC group was significantly reduced on the 14th day post-treatment. This may be due to the presence of AgNPs in the hydrogel, which have a certain anti-inflammatory effect, reducing the inflammatory response of infected wounds and significantly speeding up the wound healing process [56,57]. Moreover, AgNPs also could reduce the tumor necrosis factor receptor 1 (TNFR1) expressing on the cell surface, which could reduce the expression of TNF-α [58]. All these results indicated the excellent antibacterial, anti-inflammatory and promoting healing effect of the Ag@ODex/HA-ADH/HACC hydrogel.

Western blot analysis was used to further quantify the protein expression of IL-6, IL-1β and TNF-α. The Western blot (Figure 8A) indicated that inflammatory factors IL-6, IL-1β and TNF-α of Ag@ODex/HA-ADH/HACC had a lower expression 7 days post-operation, further induced the macrophages to generate more growth factors to facilitate the wound repairing, while caused adverse effect when the level is high. While the group control, Aquacel Ag, ODex/HA-ADH and ODex/HA-ADH/HACC all showed high expression of IL-6, IL-1β and TNF-α 7 days post-operation, indicating Ag@ODex/HA-ADH/HACC could restrain the inflammation. The protein expressions of IL-6, IL-1β and TNF-α 14 days post-operation were lower than the protein expressions 7 days post-operation (Figure 8B), and Ag@ODex/HA-ADH/HACC still exhibited the lowest expression in all groups. In conclusion, Ag@ODex/HA-ADH/HACC can enhance the wound healing through inhibiting inflammation.

## 3. Materials and Methods

### 3.1. Materials and Reagents

Dextran (Dex; Mw −70 kDa) was purchased from Macklin Co. Ltd. (Shanghai, China). Sodium hyaluronic acid (HA; Mw = 200 kDa) was obtained from Shandong Freda Biochem Co., Ltd. (Linyi, China). Chitosan quaternary ammonium salt (HACC, 90% degree of substitution) was obtained from Guangdong Wengjiang Chemical Reagent Co. Ltd. (Shaoguan, China). Hyaluronidase from purified bovine testes, silver nitrate, adipic dihydrazide (ADH) and hydroxy-benzotriazole (HOBt) were purchased from Aladdin Biochemical Technology Co. Ltd. (Shanghai, China). 1-ethyl-3-[3-(dimethylamino) propyl] carbodiimide (EDC) and sodium periodate were purchased from Guangzhou Chemical Reagent Factory. CCK-8 was purchased from Shanghai Beyotime Biotechnology Co. Ltd. (Shanghai, China) *E. coli* (ATCC8739), *S. aureus* (ATCC14458) and *P. aeruginosa* (CMCCB10104) were obtained from the Department of Biomedical Engineering of Jinan University and were maintained on solid agar medium at 4 °C. All other reagents were analytical grade and used without further processing.

### 3.2. Synthesis of Oxidized Dextran (ODex)

ODex was obtain by the oxidation of dextran according to the previous literature with a slight change. The specific method was as follows: 1 g of dextran was dissolved in pure water to prepare a 10% (*w*/*v*) aqueous dextran solution. Then, 2 mL of the concentration of 100 mg/mL of sodium periodate was added. The oxidation reaction was performed for 24 h in a dark environment. Subsequently, excess sodium periodate was removed by adding the ethylene glycol solution and stopped further oxidation. Then, the reaction solution was poured into a cellulose dialysis bag (with a molecular weight cut-off of 7–12 kDa), and dialyzed against deionized water for 72 h. The dialyzed liquor was finally lyophilized to collect the ODex powder.

### 3.3. Synthesis of HA-ADH

The modification of hyaluronic acid (HA) was performed according to the previous literature. HA (100 mg) was put into 25 mL of the deionized water, a further 800 mg of adipic dihydrazide (ADH) was added into the above liquor and stirred for 4 h to gain a clear solution (HA-ADH intermediate), and the pH was adjusted to 4.7 for further use. A total of 132 mg of HoBt and 100 mg of EDC were added into 2 mL of the DMSO/water solution (v:v = 1:1) and added to the HA-ADH intermediate solution. After that, the pH of previous liquor was maintained at 6.8 and allowed to react for another 4 h. The reaction was terminated by adjusting the pH of the solution to 7. The mixed reaction solution was dialyzed with a cellulose dialysis bag (Mw = 3500 Da) against a 100 mM NaCl solution, a 25% ethanol solution and deionized water for one day, and the dialysate was changed three times a day. HA-ADH was finally gained by lyophilization and reserved at −20 °C until use.

### 3.4. Preparation of the Hydrogel

#### 3.4.1. Preparation of ODex/HA-ADH Hydrogel

The ODex/HA-ADH hydrogel was formed through Schiff-base reaction. Specifically, 0.3 mL of 6% (*w*/*v*) ODex liquor and 0.6 mL of 2% (*w*/*v*) HA-ADH liquor were prepared and mixed homogeneously. After that, the mixture was left alone for at least 20 min until the hydrogel formed.

#### 3.4.2. Preparation of ODex/HA-ADH/HACC Hydrogel

The ODex/HA-ADH/HACC hydrogel was prepared by adding an HACC solution to the HA/ODex solution before the hydrogel formed. Specifically, 0.6 mL of 2% (*w*/*v*) HA-ADH liquor was poured into 0.3 mL of 6% (*w*/*v*) ODex liquor, then 0.3 mL of 2% (*w*/*v*) HACC liquor was added immediately and mixed homogeneously. After that, the mixture was left alone for at least 30 min until the hydrogel formed.

#### 3.4.3. Preparation of Ag@ODex/HA-ADH/HACC Hydrogel

AgNPs were obtained by the reduction of a lone pair of chitosan electrons. The ODex/HA-ADH/HACC hydrogel was fully soaked in a 0.1 M silver nitrate solution and placed in a fixed temperature incubator for 8 h at 37 °C to reduce silver ions to AgNPs.

### 3.5. Chemical Structure Characterization of the Hydrogels and AgNPs

The chemical constitution of ODex, HA-ADH, ODex/HA-ADH, ODex/HA-ADH/HACC and Ag@ODex/HA-ADH/HACC were characterized using Fourier transform infrared spectroscopy (FTIR, Bruker, VERTEX 70). KBr tablet was used as background. The scanning range was from 400 to 4000 cm^−1^, and the resolution was 4 cm^−1^. Scanning electron microscope (SEM, Philips, Philips XL-30) can detect the morphology of the lyophilized hydrogel. The ODex/HA-ADH, ODex/HA-ADH/HACC and Ag@ODex/HA-ADH/HACC hydrogels were lyophilized and coated with gold before being observed. ImageJ software was used to calculate porosity. The Ag@ODex/HA-ADH/HACC hydrogel was immersed in deionized water and centrifuged at 1000 rpm for 10 min. Then, the supernatant was taken, and the UV absorption spectrum was measured using a UV–visible spectrophotometer (Shimadzu, UV-2550) with a scanning range of 200–800 nm. The diameter of the AgNPs was characterized using a nano laser particle size analyzer (Malvern, Zetasizer Nano ZS).

### 3.6. In Vitro Swelling and Degradation Behavior of the Hydrogel

#### 3.6.1. In Vitro Swelling Behavior

The swelling performance of hydrogel is evaluated based on the gravimetric analysis. The prepared ODex/HA-ADH cylindrical hydrogels were placed in pure water and a PBS solution (pH = 7.4), respectively at 37 °C. The samples were taken from the purified water and PBS solution at a pre-selected time point. The weight of the hydrogel was recorded after carefully removing the surface water with a filter paper. The swelling ratio was evaluated using the following formula:Swelling ratio = W_t_/W_0_
where W_0_ and W_t_ represent the initial weight and the swelling weight at different time points of the sample, respectively.

#### 3.6.2. In Vitro Degradation Behavior

The prepared cylindrical shapes of ODex/HA-ADH and ODex/HA-ADH/HACC were placed in a PBS solution (pH = 7.4) and 100 U/mL of a hyaluronidase solution, respectively, at 37 °C. The PBS solution and the hyaluronidase solution were changed daily. The samples were taken from the PBS solution and the hyaluronidase solution at a pre-selected time point. The weight of the sample was recorded after carefully removing the surface water with a filter paper. The degradation ratio was assessed using the following formula:Degradation ratio = (W_D_ − W_d_)/W_D_ × 100%
where W_D_ and W_d_ represent the initial weight and the weight after degradation at different time points of the sample, respectively.

### 3.7. Rheological Property of the Hydrogel

The rheological properties of the ODex/HA-ADH, ODex/HA-ADH/HACC and Ag@ODex/HA-ADH hydrogels were measured using a rotary rheometer (Kinexus Pro+, Malvern, UK) at room temperature, and time sweep was from 0 to 300 s with a strain of 1%.

### 3.8. In Vitro Antibacterial Property

The Kirby–Bauer (KB) method was used to test the antibacterial ability against *S. aureus*, *E. coli* and *P. aeruginosa*. The prepared sample was sterilized by ultraviolet irradiation for 30 m. The solid LB medium was dripped 100 μL (1 × 10^8^ CFU/mL) of the above-mentioned bacteria suspension. Sterilized hydrogels with diameters of 10 mm were placed onto the surface of the agar. After the samples were set upright on the agar plate for 15 min, the Petri dishes were incubated for 24 h at 37 °C. The diameter of the inhibition zone on the agar plate was recorded. Each group contained three parallel samples.

### 3.9. In Vitro Cytotoxicity of the Hydrogel

A cell counting kit-8 assay was performed to analyze the cytotoxicity of hydrogels. L929 cells were applied to assess the cytotoxicity. The hydrogel was immersed in 75% alcohol for 30 min along with UV light irradiation, and then washed three times with PBS to obtain the sterilized sample. Then, the sterilized hydrogel was immersed in DMEM (10% fetal calf serum, 100 IU/mL penicillin, 100 μg/mL streptomycin) for 24 h to gain an extract, and a 10-fold dilution was prepared at the same time. To a 96-well plate, 100 μL of L929 cell suspension with a cell density of 1 × 10^5^ was added, and 100 μL of the extracts and 10-fold dilutions were added, respectively. The 96-well plate was incubated at 37 °C with a 5% CO^2^ atmosphere and the culture medium was changed every 2 days. After being cultured for 1, 3 and 7 days, the cultured medium was changed to a 10% CCK-8 solution and incubated for another 1 h at 37 °C. The microplate reader detected the OD values at 450 nm. The cell viability was assessed using the following formula:Cell viability (%) = OD(450) − OD(Control)/OD(Blank) − OD(Control)

### 3.10. In Vivo Wound Healing

Animal experiments were conducted according to the regulations of the Animal Research Center of Jinan University, which had been approved by relevant departments in advance. In this study, a total of 25 male SD rats weighing 200–250 g and aged 3 weeks old were selected as the experimental animals. All rats were anesthetized with sodium pentobarbital at a dose of 30 mg/kg and their back hair was shaved. In accordance with the previous research basis of the research group, the following burn infection model of rats was established: a hot copper table would be used to scald the back of the rat (diameter 12 mm, 90 °C) for about 30s, and the scorch formed was excised after being left for 1 day. Then, 100 μL of *P. aeruginosa* suspension (1 × 10^−8^ CFU) was used to infect the wound for 4 h. The test samples, ODex/HA-ADH, ODex/HA-ADH/HACC and Ag@ODex/HA-ADH/HACC, were sterilized by UV irradiation and then affixed to the wound. At the same time, a commercial silver ion antibacterial dressing (Aquacel silver-containing dressing) was used as a positive control group, and wounds without any scaffolds were used as a blank control group. The upper surface of all dressings was covered with a transparent silicone film to isolate moisture and bacteria from the air. Changes were recorded on the 3rd, 7th, 14th and 21st days and wounds were measured. The wound area was recorded and analyzed using ImageJ software, and the percentage of the wound area was assessed in accordance with the following formula:Wound area (%) = (W_t_/W_0_) × 100%
where W_0_ represents the wound area on day 0, and W_t_ represents the wound area on days 3, 7, 10, 14 and 21, respectively.

The excised skin tissue was homogenized in 2 mL of a 0.9% sterile saline solution on the 3rd day. Then, the homogenate was serially diluted, and 100 μL of bacterial suspension was evenly plated on selected medium agar plates.

### 3.11. Histological Analysis, Immunohistochemistry, and Western Blot

On the 3rd, 7th, 14th and 21st days, the wound repairing tissue was excised, and the sample was fixed with 4% formaldehyde in a PBS solution at 4 °C, dehydrated with a gradient concentration of ethanol and then embedded in paraffin (5 μm) and sliced. Hematoxylin-eosin (H&E) and Masson’s trichrome staining were used for staining, and the repair effect was evaluated by observing the stained sections through an optical microscope.

For immunohistochemistry, paraffinic skin sections were incubated with primary antibodies to IL-6, IL-1β and TNF-α. All steps of immunohistochemistry staining were performed according to manufacturer’s protocol. The samples of immunohistochemistry were observed under a fluorescence microscope (Zeiss, Axio Observer D1, Jena, Germany).

Samples of wound skin were completely homogenized in lysis buffer (PBS, pH 7.4), and followed by centrifugation at 10,000 rpm for 10 min. The 10% sodium dodecyl sulfate (SDS)-polyacrylamide gels were used to electrophorese the prepared proteins. Primary anti-bodies were incubated with proteins overnight at 4 °C after the proteins were transferred to PVDF Western blot membranes for 2 h at 40 V. Next, the HRP-conjugated anti secondary antibody would be applied to incubate the membrane for 1 h at 22 °C. Then, the membrane was observed via enhanced chemiluminescent reagent, and exposed to X-ray films.

### 3.12. Statistical Analysis

The data in this study were expressed as mean ± standard deviation (SD), and SPSS software (version 16.0; SPSS, Chicago, IL, USA) was used for statistical analysis. All experiments set at least three parallel samples (*n* = 3), * means *p* < 0.05, ** means *p* < 0.01 and a difference of *p* < 0.05 was considered statistically significant.

## 4. Conclusions

In this study, an ODex/HA-ADH/HACC hydrogel with AgNPs grown in situ were prepared for burn wound treating. The Ag@ODex/HA-ADH/HACC possessed a suitable swelling property, stable rheological behavior and suitable degradation rate, indicating that Ag@ODex/HA-ADH/HACC could be applied in burn wound treatment in clinical applications. Additionally, the antibacterial behavior of a synergistic antibacterial hydrogel composed of inorganic and organic antibacterial agents was analyzed, indicating that the hydrogel has a good antibacterial activity against *E. coli*, *S. aureus* and *P. aeruginosa*. The Ag@ODex/HA-ADH/HACC hydrogel-treated wound had a fast healing speed and well-organized collagen deposition. Moreover, the level of pro-inflammatory cytokines (IL-6, IL-1β and TNF-α) expressing was drastically suppressed, which may be the reason for the enhanced wound healing. In brief, the Ag@ODex/HA-ADH/HACC hydrogel could be an expectant candidate for enhancing burn wound healing.

## Data Availability

All data included in this study are available upon request by contact with the corresponding author.

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
