# Peer review of "Preparation and Application of Quaternized Chitosan- and AgNPs-Base Synergistic Antibacterial Hydrogel for Burn Wound Healing"

_molecules, 2021, doi:10.3390/molecules26134037_

Round 1
Reviewer 1 Report
The study followed the biolgical activity of a new hydrogel material, usin in vivo and in vitro experiments. The bacteriology part needs major revisions.
The methodology in chap. 3.8 is confusing and missing details. What was the concentration of bacterial inoculum? Where is the hydrogel placed? Why count the number of colonies?, because for a good Kirby-Bauer method, there should be a fine layer of confluent bacterial growth on the surface. The right terms would be "were incubated for 24h", repectively "inhibition zone were recorded".
"Each group contains three parallel samples at least" - was the experiment performed in triplicate?
Please recheck the inhibition zones diameter. From Figure 4 I deducted that paper disks with hydrogel were used. A paper disk has 6 mm, so the diameters of the inhibition zones cannot have the presented values. E. coli is misspelled in Figure 4.
Were bacteriological samples taken from the wound at least at day 3 to prove the infection?
Immunohistochemistry and Western-blot methodology is missing.
Bacterial names - with italics allover the text.
English (syntax, tenses) need revision.
Author Response
Reviewer 1:
The study followed the biological activity of a new hydrogel material, using in vivo and in vitro experiments. The bacteriology part needs major revisions.
- The methodology in chap. 3.8 is confusing and missing details. What was the concentration of bacterial inoculum? Where is the hydrogel placed? Why count the number of colonies?, because for a good Kirby-Bauer method, there should be a fine layer of confluent bacterial growth on the surface. The right terms would be "were incubated for 24h", repectively" inhibition zone were recorded".
"Each group contains three parallel samples at least" - was the experiment performed in triplicate?
Response: We deeply appreciate your constructive comments and suggestions. The details of the bacteriology part have been supplemented and revised. Kirby-Bauer (KB) method was used to test the antibacterial ability against S. aureus, E. coli and P. aeruginosa. The prepared sample was sterilized by ultraviolet irradiation for 30 min. The solid LB medium was dripped 100 μL (1×108 CFU/mL) of the above-mentioned bacteria suspension. Sterilized hydrogels with the diameter of 10 mm were placed onto the surface of the agar plate. After the samples being set upright on the agar plate for 15 min, the petri dishes were incubated for 24 h at 37 °C, respectively. Inhibition zone on the agar plate were recorded. The experiment was performed in triplicate.
- Please recheck the inhibition zones diameter. From Figure 4 I deducted that paper disks with hydrogel were used. A paper disk has 6 mm, so the diameters of the inhibition zones cannot have the presented values. E. coli is misspelled in Figure 4.
Response: We deeply thanks your comments. The diameter of the hydrogels was 10 mm, so the diameters of the inhibition zones can have the presented values, and the scale bar has been added in the manuscript.
- Were bacteriological samples taken from the wound at least at day 3 to prove the infection?
Response: The excising skin tissue were homogenized in 2 mL of 0.9% sterile saline solution on 3th day. Then the homogenates were serially diluted, and 100 µL bacterial suspension was evenly plated on selection medium agar plates. The content of bacteria as shown in Figure 5 (C), which revealed the wounds were successfully infected by P. aeruginosa, the group of Ag@ODex/HA-ADH/HACC has the excellent antibacterial activity in vivo.
- Immunohistochemistry and Western-blot methodology is missing.
Response: The detailed information of the immunohistochemistry and western-blot methodology has been added in the part 3.11 of the manuscript. The newly added contents have been marked in the revised manuscript and detailed contents are as follows:
For immunohistochemistry, paraffinic skin sections were incubated with primary antibodies to IL-6, IL-1β and TNF-α. All steps of immunohistochemistry staining were according to manufacturer's protocol. The samples of immunohistochemistry were observed under a fluorescence microscope (Zeiss, Axio Observer D1, Germany).
Samples of wound skin were completely homogenized in lysis buffer (PBS, pH 7.4), and followed by centrifugation at 10,000 rpm for 10 min. The 10% sodium dodecyl sulfate (SDS)-polyacrylamide gels were used to electrophoresed prepared proteins. Primary anti-bodies were incubated with proteins overnight at 4°C after the proteins transferring to PVDF western blot membranes for 2 h at 40 V. Next the HRP-conjugated anti secondary antibody would be applied to incubate the membrane for 1 h at 22°C. Then the membrane was observed via enhanced chemiluminescent reagent, and exposing to X-ray films.
- Bacterial names - with italics allover the text.
Response: The bacterial name has been revised with italics all over the text.
- English (syntax, tenses) need revision.
Response: The revised manuscript has been polished by native English speaker.

Reviewer 2 Report
The preparation of antibacterial materials based on chemically modified polysaccharides and silver nanoparticles, the study of their properties constitutes the substantial part of the presented manuscript. Research of this kind is currently an actively developing area at the intersection of chemistry, microbiology, and medicine. The manuscript contains a description of the methods for the preparation of initial hydrogels, in situ synthesis of silver nanoparticles, and a study of the antibacterial properties of the obtained materials. The authors have done a significant amount of work, the conclusions are based on the results of experiments.
However, there are comments on the text presented.
- The introduction lacks information on the mechanisms of the antibacterial effect of silver nanoparticles.This effect may be due to both the nanoparticles themselves and the products of their oxidation under aerobic conditions.
- The antimicrobial effect of both chitosan and silver nanoparticles is known.Quite often, a synergistic effect occurs when both of these components are simultaneously contained in the system.In the introduction, it is necessary to note this fact and confirm it with literary references.
- The authors describe the formation of Schiff bases.It is known that these compounds are hydrolytically unstable.This can be captured using spectral methods.Is there a long-term change in the spectral properties of the synthesized hydrogels?
- The value of the standard electrode potential of the reduction reaction of the silver ion allows the oxidation of the hydroxyl groups of polysaccharides to carbonyl.Has this process been investigated?
- The particle size distribution curves obtained by the DLS method (Fig. 1C) can correspond not only to silver nanoparticles, but also to sparingly soluble polysaccharides.A similar curve of the hydrogel without silver nanoparticles should be presented.
- The absorption maximum of silver nanoparticles (Fig. 1B) is at 417 nm.In the text (page 4 of the manuscript) 471 nm is indicated. A correction needs to be made.
After making corrections, the manuscript can be published in the "Nanomaterials"
Author Response
Reviewer 2:
The preparation of antibacterial materials based on chemically modified polysaccharides and silver nanoparticles, the study of their properties constitutes the substantial part of the presented manuscript. Research of this kind is currently an actively developing area at the intersection of chemistry, microbiology, and medicine. The manuscript contains a description of the methods for the preparation of initial hydrogels, in situ synthesis of silver nanoparticles, and a study of the antibacterial properties of the obtained materials. The authors have done a significant amount of work, the conclusions are based on the results of experiments.
However, there are comments on the text presented.
The preparation of antibacterial materials based on chemically modified polysaccharides and silver nanoparticles; the study of their properties constitutes the substantial part of the presented manuscript. Research of this kind is currently an actively developing area at the intersection of chemistry, microbiology, and medicine. The manuscript contains a description of the methods for the preparation of initial hydrogels, in situ synthesis of silver nanoparticles, and a study of the antibacterial properties of the obtained materials. The authors have done a significant amount of work, the conclusions are based on the results of experiments.
However, there are comments on the text presented.
- The introduction lacks information on the mechanisms of the antibacterial effect of silver nanoparticles. This effect may be due to both the nanoparticles themselves and the products of their oxidation under aerobic conditions.
Response: We deeply appreciate your constructive comments. We have added the mechanisms of the antibacterial effect of silver nanoparticles in introduction in the revised manuscript and marked it in blue.
Diverse types of inorganic particles such as metal, metal oxides, non-metal oxides and others have been applied as an antibacterial agent. Ag nanoparticles (AgNPs) is one of the inorganic antibacterial agents that extensive used in biomedical area. AgNPs have an extensive spectrum of biocidal and antibacterial activities, mainly because they can damage the respiratory system and transmission system in bacterial cells, thereby inhibiting bacterial metabolism and hindering the reproduction of bacterial DNA. Furthermore, they can also disrupt the scavenging mechanisms by directly binding with thiol groups in related enzymes and GSH into its oxidized form, glutathione disulfide (GSSG), which results in increased concentrations of ROS and free radicals. and further direct damage the bacteria.
- The antimicrobial effect of both chitosan and silver nanoparticles is known. Quite often, a synergistic effect occurs when both of these components are simultaneously contained in the system. In the introduction, it is necessary to note this fact and confirm it with literary references.
Response: The synergistic effect of silver nanoparticles and quaternized chitosan has been mention in the manuscript, and the references also has been noted.
Previous studies have shown that quaternary ammonium chitosan and silver nano-particles have synergistic effects in antibacterial property, but the hydrogels that combine with these two complexes with a rarely report.
- The authors describe the formation of Schiff bases. It is known that these compounds are hydrolytically unstable. This can be captured using spectral methods. Is there a long-term change in the spectral properties of the synthesized hydrogels?
Response: The degradation of the hydrogels also could capture the long-term change, which can prove the compounds with Schiff bases are hydrolytically unstable. For the in vivo use of biomaterial, the degradability is very important. In the PBS liquor, the degradation is mainly based on the hydrolytic scission of molecular chains, and broken of the amide bond.
- The value of the standard electrode potential of the reduction reaction of the silver ion allows the oxidation of the hydroxyl groups of polysaccharides to carbonyl. Has this process been investigated?
Response: Silver ion allows the oxidation of the hydroxyl groups of polysaccharides to carbonyl. However, in our system, the ODex also has the group of carbonyl, which could interfere the investigation of the process that silver ion allows the oxidation of the hydroxyl groups of polysaccharides to carbonyl. Thus, further determination was not conducted.
- The particle size distribution curves obtained by the DLS method (Fig. 1C) can correspond not only to silver nanoparticles, but also to sparingly soluble polysaccharides. A similar curve of the hydrogel without silver nanoparticles should be presented.
Response: The DLS without AgNPs has been added in the supporting information Figure S1. Figure A exhibited the size of the hydrogel without silver nanoparticles, the size is greater than 190 nm and within the scope of 190-550 nm. While the Figure B exhibited the size distribution of silver nanoparticles, the size of hydrogel with silver nanoparticles were mostly (>50%) within the scope of 60-200 nm, which could not be observed in the Figure A. To sum up, we could infer that the silver nanoparticles were within the scope of 60-190 nm.
- The absorption maximum of silver nanoparticles (Fig. 1B) is at 417 nm. In the text (page 4 of the manuscript) 471 nm is indicated. A correction needs to be made.
Response: We deeply appreciate your constructive comments and suggestions. We have revised absorption maximum of silver nanoparticles is at 417 nm at the page 4 of the manuscript.
After making corrections, the manuscript can be published in the "Nanomaterials"

Round 2
Reviewer 1 Report
The authors addressed almost all important issues.
Nevertheless, Figure 4 still needs correction. Considering the scale and the disk diameter, the presented values of the inhibition zones are not the correct ones. For example the 6 mm refers probably to the inhibition "ring" width, not zone diameter. The diameter is measured as the distance between popsite edges of the outer ring. Considering a 10 mm disk, the vale should be 22 mm (6+10+6). Or instead of 14, should be 38 (14+10+14); and so on...
Also Figure 4(B) of course needs numerical correction
See page 17 from this protocol for example https://asm.org/getattachment/2594ce26-bd44-47f6-8287-0657aa9185ad/Kirby-Bauer-Disk-Diffusion-Susceptibility-Test-Protocol-pdf.pdf
Line 438: "The excising skin tissue were homogenized in 2 mL 0.9% sterile saline solution on 3th day. Then the homogenates were serially diluted, and 100 μL bacterial suspension was evenly plated on selection medium agar plates" corrected to "The excised skin tissue was homogenized in 2 mL 0.9% sterile saline solution on 3th day. Then the homogenate was serially diluted, and 100 μL bacterial suspension was evenly plated on selection medium agar plates"
Author Response
The authors addressed almost all important issues.
Nevertheless, Figure 4 still needs correction. Considering the scale and the disk diameter, the presented values of the inhibition zones are not the correct ones. For example the 6 mm refers probably to the inhibition "ring" width, not zone diameter. The diameter is measured as the distance between popsite edges of the outer ring. Considering a 10 mm disk, the vale should be 22 mm (6+10+6). Or instead of 14, should be 38 (14+10+14); and so on...
Also Figure 4(B) of course needs numerical correction
See page 17 from this protocol for example https://asm.org/getattachment/2594ce26-bd44-47f6-8287-0657aa9185ad/Kirby-Bauer-Disk-Diffusion-Susceptibility-Test-Protocol-pdf.pdf
Response: We deeply thanks your comments. The diameter of the inhibition zones has been revised in the manuscript.
Line 438: "The excising skin tissue were homogenized in 2 mL 0.9% sterile saline solution on 3th day. Then the homogenates were serially diluted, and 100 μL bacterial suspension was evenly plated on selection medium agar plates" corrected to "The excised skin tissue was homogenized in 2 mL 0.9% sterile saline solution on 3th day. Then the homogenate was serially diluted, and 100 μL bacterial suspension was evenly plated on selection medium agar plates"
Response: We have revised as suggested.
